# Immunoglobulins with Non-Canonical Functions in Inflammatory and Autoimmune Disease States

**DOI:** 10.3390/ijms21155392

**Published:** 2020-07-29

**Authors:** Evgeny A. Ermakov, Georgy A. Nevinsky, Valentina N. Buneva

**Affiliations:** 1Institute of Chemical Biology and Fundamental Medicine, Siberian Branch of the Russian Academy of Sciences, 630090 Novosibirsk, Russia; evgeny_ermakov@mail.ru (E.A.E.); nevinsky@niboch.nsc.ru (G.A.N.); 2Novosibirsk State University, Department of Natural Sciences, 630090 Novosibirsk, Russia

**Keywords:** canonical, non-canonical functions of immunoglobulins, catalytic antibodies, abzymes, inflammatory, autoimmune diseases

## Abstract

Immunoglobulins are known to combine various effector mechanisms of the adaptive and the innate immune system. Classical immunoglobulin functions are associated with antigen recognition and the initiation of innate immune responses. However, in addition to classical functions, antibodies exhibit a variety of non-canonical functions related to the destruction of various pathogens due to catalytic activity and cofactor effects, the action of antibodies as agonists/antagonists of various receptors, the control of bacterial diversity of the intestine, etc. Canonical and non-canonical functions reflect the extreme human antibody repertoire and the variety of antibody types generated in the organism: antigen-specific, natural, polyreactive, broadly neutralizing, homophilic, bispecific and catalytic. The therapeutic effects of intravenous immunoglobulins (IVIg) are associated with both the canonical and non-canonical functions of antibodies. In this review, catalytic antibodies will be considered in more detail, since their formation is associated with inflammatory and autoimmune diseases. We will systematically summarize the diversity of catalytic antibodies in normal and pathological conditions. Translational perspectives of knowledge about natural antibodies for IVIg therapy will be also discussed.

## 1. Introduction

Immunoglobulins (Igs) are involved in numerous molecular mechanisms of both innate and adaptive immune systems due to the presence of two functional centers (antigen binding sites (Fab) and fragment crystallizable region (Fc)), as well as a combination of their unique structural and functional properties. Generally, Igs combine the Fab domain-mediated antigen recognition process with the activation of various innate responses mediated by the involvement of various receptors, protective proteins and immune cells by the Fc domain [1]. For a long time, it was believed that the host’s immune system generates only highly specific antibodies (Abs) against pathogens [2]. However, the development of experimental methods and animal models made it possible to determine the extreme heterogeneity of the Igs pool, reflecting a wide variety of biological functions of Abs.

Igs have been shown to exhibit their effector functions both in normal and pathological conditions, especially in inflammatory and autoimmune diseases (AIDs). The pathological role of Abs is well documented in the literature (e.g., [3]). At the same time, natural Igs are powerful immunomodulators that both induce and suppress immune responses and inflammatory processes [4]. However, much less attention is paid to the study of Ab effector functions beyond the classical concepts of immunology. Evidence of a significant functional diversity of Abs is the various identified therapeutic effects of intravenous Igs (IVIg) [5]. IVIg preparations are widely used to treat various immunological pathologies. A better understanding of these Ab functions can enrich existing therapeutic strategies.

In this review, we circumstantially describe recent findings concerning the canonical and non-canonical functions of Igs. We discuss the broad diversity of naturally occurring Abs that explain the wide variety of Ab functions. In addition, we pay special attention to catalytic Abs, because they are associated with autoimmune and inflammatory disease states. For the first time, we systematically summarize the diversity of catalytic Abs in normal and pathological conditions. Translational perspectives of knowledge about natural Abs for IVIg therapy are also considered.

## 2. Canonical and Non-Canonical Functions of Immunoglobulins

The human Ab repertoire is created by somatic evolution in B cell populations, which allows the immune system to recognize and eradicate almost any antigen [6]. State-of-the-art immune repertoire next-generation sequencing technologies have allowed a deeper understanding of the diversity of Abs and B cell receptors and thus the immune status of the individuals [7,8,9]. The Ab repertoire is flexible and changeable throughout life. Flexibility is achieved due to the presence of an extensive repertoire of native Abs, the diversity of which is increased by somatic hypermutation after exposure to the antigen. Each of the approximately estimated 5 × 10^9^ B cells produces a specific B cell receptor or Ab through somatic recombination of variable (V), diversity (D), joining (J) and constant (C) gene segments (V–(D)–J recombination) [8,9]. The V–(D)–J recombination process results in a light (L) chain, assembled from V, J and C gene segments from one of the light chain loci, and a heavy (H) chain, assembled from V, D, J and C gene segments from the heavy chain locus [8]. Pairs of H and L chains combine to form the following Ab isotypes: monomeric IgG, IgE and IgD; dimeric IgA; and pentameric IgM. Fab contains the variable segments of H and L chains (VH and VL) that bind to a specific surface on the antigen (epitope). Each of the VH and VL segments contains three complementarity-determining regions (CDRs) and four framework regions. The CDRs contain the amino acid residues responsible for antigen binding (paratope). The process of somatic hypermutation after exposure to antigens introduces mutations, primarily in CDRs [8]. Moreover, Abs have constant domains involved in the formation of Fc, which is responsible for interacting with diverse receptors or complement. Thus, a broad Abs repertoire, a unique structure and processes of affinity maturation determine the variety of functional activities of Abs.

Ab-mediated biological functions should be considered according to functional level [10]. The simplest functional level implies the elementary interaction of the Fab domain with the antigen to neutralize the pathogen. The second functional level is the initiation of secondary reactions following antigen recognition. This level is due to both the binding activity of the Fab domain and recognition of the Fc domain by innate immune receptors, complement or immune cells [10]. Examples of well-characterized Ig functions of this functional level are Ab-dependent complement activation, complement-mediated lysis of pathogens or infected cells, Ab-dependent cell-mediated cytotoxicity, phagocytosis, etc. [1,11]. The highest functional level reflects the overall biological role of the Ab in the host’s immune defense. All of these functions are ultimately aimed at protecting against pathogens and maintaining immune homeostasis [10]. However, with the development of research technologies, it has become clear that Igs’ general biological functions are not limited to antigen recognition processes and the initiation of innate immune responses. In recent years, many Ab functions have been discovered that do not fit into the classical paradigm [10].

Given the available data, Ig functions can be nominally divided into canonical and non-canonical (summarized in Table 1). The canonical functions of immunoglobulins are well documented (including but not limited to [1,11,12,13,14,15,16,17,18,19]), thus we will describe them only briefly.

A quick look at Table 1 allows us to make the corollary that Abs exhibit more numerous canonical functions at the second and highest functional levels. However, the functions of Igs should be considered depending on their classes and subclasses, since their functions differ significantly. For example, pathogen recognition by IgG triggers complement activation, Ab-dependent cell-mediated cytotoxicity, or directs immune complexes to degradation into proteasomes through endocytosis after interaction with C1q and TRIM21 [10]. In general, IgG3 has the highest functional activity, followed by IgG1, whereas IgG2 and IgG4 have the least functional activity in subclasses. IgG also provides specific protection for newborns from certain pathogens as a result of the transport of the mother’s IgG through the neonatal Fc receptor (FcRn) [14]. IgM can markedly activate complement [29]. Together with complement, IgM provides transport of antigens to secondary lymphoid organs and the initiation of an immune response [10,13,14]. Secretory IgA (sIgA) is classically known to promote both immune exclusion and immune inclusion of various microorganisms on the mucous membranes [13,14]. Moreover, sIgA neutralizes intracellular pathogen determinants in the epithelial cell endosomes. Furthermore, sIgA provides antigen release into the lumen of the mucous membrane through its secretion after the interaction of the sIgA-antigen complex with the polymeric immunoglobulin receptor (pIgR) [13,14]. Thus, IgA actively regulates the diversity of commensal bacteria in the intestine [13,14]. However, it is important to note that microbiota metabolites affect the production of IgA, as well as systemic IgG responses [22]. IgE, in its turn, initiates mast cell and basophil degranulation after antigen recognition by IgE, associated with the Fc receptor for IgE (FcεRI). IgE also mediates destruction and removal of helminths and other pathogens, as well as inactivation of animal poisons and toxins [17,18]. Secreted IgD is involved in the regulation of commensal and pathogenic bacteria and mucosal allergens [18]. Furthermore, each of the classes of Igs is involved in the regulation of proliferation, development and homeostasis of the immune system cells, depending on the receptors expressed. Indeed, IgG participates in B cell selection and survival, as well as regulation of plasma cell apoptosis [11,12]. According to recent data, IgM through Fc receptor for IgM (FcμR) is involved in the regulation of B cell development and IgG production, thereby providing immune tolerance [12,14]. Serum monomeric IgA, on the one hand, promotes anti-inflammatory response after interaction with the Fc receptor for IgA (FcαRI) and other receptors [13]. On the other hand, IgA immune complexes contribute to a pro-inflammatory response after interaction with FcαRI and pattern recognition receptors (PRRs) [14]. Thus, the realized effect depends on the joint involvement of other receptors. Furthermore, IgE regulate growth, maturation, survival and homeostasis of mast cells [17]. IgD receptors expressed on B cells regulate their development and maturation, as well as clonal anergy and self-tolerance [19,30]. IgD associated with basophils and other cells, after antigenic stimulation, triggers the release of IL-4, which causes the production of IgG by B cells [19].

Thus, most of the biological activity of various classes of Abs is mediated through interactions between the Fc domain and Fc receptors, while additionally involving other receptors [28,31]. Five Fc receptors for IgG (FcγR), two Fc receptors for IgE (FcεR), one for IgA (FcαR) and for IgM (FcμR), as well as for IgA and IgM (FcRα/μR), were identified in humans [12,16,28]. The diversity of Fc domains further extends Ig functions due to the presence of sequence polymorphisms and variations in the glycosylation patterns [32,33]. Activation of Fc receptors by Igs or immune complexes leads to several subsequent effects, depending on the Fc receptor-expressing cell, type of Ig or immune complex, cytokine environment and complement presence. Consequently, depending on the immunological context, the responses initiated are different. The diversity of canonical biological outcomes caused by different classes and subclasses of Abs allows for fine-tuning of the immune response, depending on the pathological conditions.

Non-canonical functions of Igs impart even more possibilities for immune responses. These functions include either atypical strategies for pathogen neutralization or actions distinctive of other proteins (extensively reviewed in [10]). Such non-canonical functions of Abs are often due only to theAb’s Fab domain (see Table 1). These functions arise in consequence of the extreme diversity of sequences and structural conformations of the antigen-binding site, which are formed both by genetic processes of recombination and mutation, and by post-translational mechanisms [34]. The catalytic activity of Igs is typical, and is one of the most common examples of the non-canonical function of different classes of Igs. The results of structural and sequence analyses of the antigen-binding sites of some Abs demonstrate the similarity of sequences and topographic characteristics with the active sites of canonical enzymes [20]. These features give Igs the ability to catalyze specific chemical reactions [20]. Abs with catalytic activity can promote immune defense by hydrolysis of functional molecules important for pathogens. In addition, they can minimize the autoimmune response by reducing the amount of antigen available for immune recognition, or by hydrolyzing pro-inflammatory molecules. Abs with natural catalytic activity are found in the immune repertoire in both physiological and pathological conditions. Immune pathologies are often accompanied by both the expanded diversity and increased level of catalytic activity of Igs. Given this, we will discuss catalytic Abs in more detail below.

In addition to catalytic activity, non-canonical functions include direct inactivation of pathogens in the absence of effector cells or molecules. For example, two monoclonal Abs, IgM and IgG, induced changes in the expression genes and metabolism of the *Cryptococcus neoformans* fungal pathogen after binding to the cell surface [10]. Furthermore, it was shown that the Fc domain of sIgA, interacting with bacterial glycans, can modulate the expression of polysaccharide utilization loci, including an uncharacterized family called the mucus-associated functional factor (MAFF) family [35]. Another example is a monoclonal Ab specific to the *Escherichia coli* β-barrel assembly machine (BamA), which interferes with the folding and assembly of membrane proteins [36]. Thus, some Abs are able to change the basic biological processes of pathogens, leading to their inactivation. Furthermore, several Abs can cause conformational changes in the target molecules of the pathogen [10].

Abs with agonistic activity play an important role in the expansion of the functions of Igs. Many examples are known in which Igs modulate intracellular signaling, including acting as agonists or antagonists of receptors [37]. In some cases, signaling modulation occurs through the Fc domain. For example, IgGs can modulate intracellular insulin signaling due to the interaction of the hyposialylated IgG Fc domain with FcγRIIb [38].

A no less important non-canonical function of Igs is the compensation of innate immune defects due to anti-cytokine activity or other mechanisms [10]. For example, Abs against staphylococcal lipoteichoic acid can protect against staphylococcal infections in individuals with congenital insufficiency of the toll-interleukin 1 receptor (TIR) domain-containing adapter protein (TIRAP) [27]. It is also interesting that some IgGs are involved in the carriage, bioavailability regulation and protection of hormones from proteolytic degradation [10]. However, Ab-dependent enhancement of infection or disease may be a negative function of IgG [31].

Non-canonical functions of IgA, in addition to catalytic activity, include regulation of penetration into the systemic circulation by microbial metabolites, which are involved in the regulation of the metabolism and immunity of the host [14]. Moreover, high-avidity pathogen-specific sIgA contributes to the formation of bacterial clusters, “enchained growth” and enhanced clearance [14]. IgA is also involved in the transepithelial transfer of bacteria from the small intestine to Peyer’s patches and induction of T cell-dependent Ab responses [10].

Non-canonical functions of IgM can be attributed to catalytic activity, direct inactivation of pathogens in the absence of effector cells and molecules, as well as modulation of lymphocyte intracellular signaling due to the interaction of the Fc domain of IgM with FcμR [12,28].

Thus, Igs due to canonical and non-canonical functions significantly expand the functionality of the immune system. The variety of canonical and non-canonical functions of Igs is also reflected in the extreme diversity of Abs generated in an organism.

## 3. Diversity of Abs Types: Antigen-Specific, Natural, Polyreactive, Broadly Neutralizing, Homophilic, Bispecific, Catalytic Abs

The plethora of Ig functions in the immune system may partly be explained by the wide variety of synthesized Ab types. For a long time, the central paradigm of immunology has been that the immune system generates highly specific Abs against environmental components. At the same time, it learns not to recognize the components of its tissues during ontogenesis (the clonal selection theory) [2]. According to this theory, it was also argued that the failure of immunological tolerance leads to the generation of autoantibodies (autoAbs) and the development of AIDs. The results of investigations of the presence, in healthy individuals, of autoAbs that recognize their autoantigens with low avidity were mainly ignored [2]. However, with the development of research technologies and complex animal model systems, more studies have appeared that prove that autoreactive B cells, as well as natural auto- and polyreactive Abs, are abundantly present and are actively involved in maintaining immunological homeostasis.

The immune system constantly generates many types of Abs (summarized in a somewhat simplified form in Table 2). In general, antigen-specific or adaptive Abs are generated by plasma cells in response to the antigen. During the primary immune response, antigen-presenting cells recognize the pathogen through numerous PRRs and present the processed antigen to B cells [39]. After antigen stimulation, somatic hypermutation and clonal selection, B cells become long-lived plasma cells that produce antigen-specific adaptive Abs in the secondary immune response [39]. Such Abs are characterized by high affinity and specificity. Their main role is the specific binding of the antigen and the initiation of innate responses (see Table 2). The functions of antigen-specific Abs in inflammatory and AIDs are well described [3], so we will not review them here.

The follicular or B2 B cells are known to be the main producers of antigen-specific Abs and the most common B cells in humans [39]. However, B1 cells exist in mice, as well as most likely in humans [40]. B1 lymphocytes are a special subtype of B cells producing polyreactive and low-affinity natural Abs against viral and bacterial antigens [41]. B1 lymphocytes respond effectively to non-specific inflammatory and pathogen-associated stimuli by migrating to secondary lymphoid tissues, where they undergo rapid differentiation into plasma cells secreting natural Abs [40]. B1 cells are selected for autoreactivity and form a pool of long-lived, self-renewing B cells that produce the majority of circulating, naturally occurring, low-affinity IgM and IgG, that cross-react with both autoantigens and conservative microbial antigens. According to recent data, Toll-like receptors (TLR)-mediated activation of B1 cells through CpG oligonucleotides, or lipopolysaccharides, leads to a rapid increase in the production of protective natural Abs (IgM, IgG) [42,43]. Interestingly, B2 lymphocytes can differentiate directly into B1 lymphocytes, and this process is controlled by a self-reactive B cell receptor [44]. Depending on the presence or absence of a surface CD5 marker, B1 cells can be subdivided into subpopulations of B1a (CD5^+^) and B1b (CD5^−^) [40]. It was found that B1a cells producing natural Abs counteract a wide range of viral and bacterial antigens, whereas B1b cells secrete more specific induced Abs against certain bacteria [45]. In summary, B2 cells produce highly specific Abs, whereas B1 cells synthesize non-specific, polyreactive, low-affinity Abs against various antigens.

The biological role of natural Abs (mainly IgM and IgG) produced by B1 cells is direct neutralization of specific pathogens, classical complement activation, antigen transport to secondary lymphoid organs and presentation, Ab-dependent cell-mediated cytotoxicity, phagocytosis of apoptotic cells, regulation of immune cells homeostasis, etc. (see Table 2) [46]. Ultimately, natural Abs help reduce inflammation and autoimmune reactions by removing damage-associated molecular patterns (DAMPs), such as extracellular DNA [45,46].

Among natural Abs, polyreactive Abs are also nominally distinguished (see Table 2) [4,47]. During the development of B cells, highly polyreactive Abs are known to be removed from the repertoire using physiological mechanisms, including deletion and editing of receptors. However, a low number of polyreactive Abs remains. Such Abs have pronounced protective functions against autoantigens, as well as viral pathogens [48]. Furthermore, polyreactive Abs are involved in the regulation of microbiota [49]. Among polyreactive Abs that neutralize viral pathogens (especially human immunodeficiency virus (HIV)), broadly neutralizing Abs are especially distinguished [50]. A feature of these Abs is that they are able to neutralize multiple HIV-1 clades [50]. There are high hopes associated with the development of HIV vaccines based on broadly neutralizing Abs [51].

Some Igs are capable of self-association through interactions between their antigen-binding sites [52]. Such Igs are called homophilic Abs. Homophilic interactions allow for more efficient binding to the target antigen, especially antigens with a repetitive nature. For example, highly protective Abs against specific antigens of *Plasmodium falciparum* have recently been shown to use homophilic interactions to bind more efficiently to the antigen [53]. Homophilic Ab interactions promote clustering of many Ig molecules, increased affinity for repeating antigens and subsequently more efficient recruitment of the complement system and activation of B cells [52,53]. Interestingly, some therapeutic Abs (e.g., rituximab) use homophilic binding to recognize antigens and promote cytotoxicity [52].

Bispecific Abs containing two different antigen-binding sites are most often generated by genetically engineered pathways [54]. However, such Abs can be detected in low concentrations in humans. In our work, we showed that natural bispecific Abs are found in the blood of healthy humans [55]. Nearly 9% of IgG molecules of healthy donors contained two different chains of both the kappa and lambda light chains simultaneously. Furthermore, bispecific Abs were observed in human placenta (IgG1-IgG4) [56]. It was shown that up to 15.0% IgG from placenta contained two different (kappa and lambda) light chains simultaneously. Moreover, human milk also contains chimeric kappa and lambda Abs (IgG and sIgA) in even higher concentrations than in the serum of healthy donors and the placenta [57,58].

In addition to these types of Abs, the immune system also generates catalytic Abs (see Table 2). Their formation is most often associated with autoimmune reactions. The following part of the review is devoted to the consideration of the features of catalytic Abs.

This largely simplified and possibly incomplete scheme (see Table 2) reflects the diversity of Ab types. Different types of Abs provide different functions of the adaptive immune system. We cannot exclude the possibility that other specific types of Abs possessing new canonical and non-canonical functions will be discovered.

## 4. Catalytic Immunoglobulins

Catalytic Abs, or abzymes, are immunoglobulins (Igs) of various classes, with the ability to not only bind the antigen but also catalyze chemical reactions with its participation [59]. Catalytic Abs are usually divided into natural ones, formed in an organism, and artificial ones, obtained by immunization with stable analogs of transition states of chemical reactions, or other genetic engineering methods. Nevertheless, the question of the mechanisms of formation and the role of catalytic Abs in the functioning of the immune system in normal and pathological conditions remains poorly understood.

### 4.1. Origin and Generation of Catalytic Immunoglobulins

The main mechanisms of the formation of Abs with enzymatic properties are currently well studied in in vitro experiments with monoclonal Abs. However, the possibility of realizing a particular mechanism in vivo is difficult to assess. However, there are several mechanisms for the formation of catalytic Abs (Figure 1).

The first mechanism for generating catalytic Abs is associated with genetic mechanisms. The amazing variety of Abs produced by an organism (theoretically 10^24^–10^26^ possible variants) is due to the processes of V–(D)–J recombination (V—variable, D—diversity and J—joining gene segments) and somatic hypermutation of germ genes that proceeds after antigenic stimulation [61]. Nucleophilic sites capable of catalyzing chemical reactions by the mechanism of nucleophilic catalysis of serine proteases (catalytic triad) were found in the variable domains of Abs encoded by the original conservative germline genes [62,63]. Using bioinformatics methods, it was shown that genes encoding the catalytic regions of Abs exhibit a high degree of homology with their germline counterparts [63]. Interestingly, the amino acid residues involved in catalysis are found not only in CDRs, but also in the framework regions of variable domains of Abs [63]. In addition, it has been shown that in some cases, at least one amino acid in the catalytic triad was obtained by somatic hypermutation [63]. These data indicate that the mechanisms of hypermutation for the acquisition of catalytic function may be different from those for improving affinity [63]. Therefore, the catalytic function of Abs is encoded in the germline genes. Consequently, the maturation process by the accumulation of somatic hypermutations is less critical for the catalytic function of Abs.

The second mechanism of the formation of catalytic Abs is immunization with stable transition state analogs (TSAs) (Figure 1B) [60]. The resulting Abs are complementary to this transition state and lead to a catalytic acceleration of this reaction. Using this method, the first artificial catalytic Abs that catalyze the hydrolysis of amides and esters, as well as cyclization, decarboxylation, lactonization, and redox reactions were obtained [64,65]. Moreover, abzymes for which there are no analogs among natural enzymes are described, and abzymes that require cofactors similar to classical enzymes are obtained [59].

The third mechanism of the formation of catalytic Abs is “reactive immunization”, with antigens conjugated to highly reactive haptens (Figure 1C), which leads to the formation of a covalent complex with the B cell receptor (BCR) and the induction of adaptive maturation of catalytic Abs with nucleophilic sites [66]. Abzymes that hydrolyze the gp120 protein of the HIV-1 virus envelope [67] and that hydrolyze β-amyloid [68] were obtained by this approach.

The fourth mechanism of abzyme formation is based on the theory of the idiotypic–anti-idiotypic network postulated by Niels Jerne in 1974 (Figure 1D) [69]. According to this theory, immunization with an antigen results in the formation of 1st generation Abs (Ab1) with an antigen-binding site specific for the antigen epitope. Immunization with the variable region Ab1 induces the formation of Abs of the 2nd generation (Ab2), the antigen-binding domain of which is complementary to the variable region Ab1. In some cases, this antigen-binding domain is an internal image of the antigenic epitope. If the initial antigen is an enzyme, then some of the Ab2 will be analogous to the active center of the enzyme and will be catalytically active (Figure 1D). Using this method, monoclonal abzymes with acetylcholinesterase [70], lactamase [71] and proteolytic [72] activities were obtained and studied.

Standard immunization with various antigens, including conjugated with adjuvants, can lead to catalytic Abs. Large molecules (DNA, RNA, proteins) [73,74,75,76], and haptens (peptides, oligosaccharides, chemical compounds) [77,78] can act as immunogens. Antigens of the body’s tissues can also act as immunogens. Some molecules can change their conformation and acquire immunogenic properties due to various destructive processes, such as inflammation, oxidative stress, exposure to toxic substances and environmental factors [79]. Apoptotic cell death can also be a source of autoantigens [80]. When interacting with proteins, some antigens can take on a conformation similar to the structures of transition states of chemical reactions. Consequently, the formation of catalytic Abs in a living organism can be carried out by all of the following mechanisms: due to the adaptive maturation of nucleophilic centers of Abs originating from the original germline genes, due to the formation of Abs to analogs of transition states, due to immunization with reactive analogs and due to the idiotypic–anti-idiotypic interactions of Abs (Figure 1).

The origin of plasma cells producing catalytic Igs is not well understood. However, in recent studies conducted by our laboratory on mouse models, it was shown that the formation of catalytic Abs is associated with disturbances of the differentiation profile of bone marrow hematopoietic stem cells. [81,82,83]. The generation of catalytic Abs is known to be limited by B cell maturation [4]. Upon repeated encounters with the antigen, its interaction with the B cell receptor (BCR) should lead to the activation of B cell proliferation. The rapid degradation of the antigen in the case of catalytic activity in the BCR deprives B cells of the stimulation necessary for proliferation. However, in conditions of excess antigen, this mechanism is of lesser importance. In general, the rate of catalysis demonstrated by Abs is limited by the rate of stimulation of the BCR signal to induce proliferation [4]. In this case, disturbances of signaling through the BCR can lead to the development of AIDs [84], which in turn will increase the production of catalytic Abs [85]. The selection mechanisms of B cell clones producing catalytic Abs need to be investigated in detail.

Thus, catalytic Abs are an important component of the total pool of Igs. The total pool of Abs consists not only of high-affinity, antigen-specific Abs, but also of catalytic Abs with less affinity and specificity. Moreover, it also contains natural, polyreactive, broadly neutralizing, homophilic, bispecific Abs, etc. (see Section 3). This variety of Ab types reflects the breadth of the Ab repertoire. However, it is quite challenging to estimate the amount of catalytic Abs formed, since it depends on the individual characteristics of an organism, the nature of the immunogen that caused the formation of Abs and other reasons. An analysis of the presence of catalytic motifs of serine proteases among known structures of Abs in the Protein Data Bank showed that 7.4% of Abs have motifs of a serine protease-like triad [86]. However, not all Abs having a catalytic triad motif exhibit catalytic activity. Perhaps this is due to the rigid structure of the Ab molecule, which prevents the amino acids of the catalytic triad from converging to form the catalytic center [87]. In addition, it was shown that constant domains play an essential role in the regulation of the catalytic activity of Abs. In particular, when coexpressing a variable domain with an IgM constant domain, proteolytic activity was much higher than when coexpressing with an IgG constant domain [88]. Nevertheless, the induction of the formation of catalytic Abs occurs in pathological conditions, especially in diseases associated with disorders in the immune system.

### 4.2. Catalytic Abs in Normal and Pathological Conditions

The information currently known (as of May 2020) about natural catalytic Abs generated in the body is presented in Table 3. This table only includes data on the catalytic activity and substrate specificity of Igs of various classes (IgG, IgM, sIgA) found in blood serum, cerebrospinal fluid (CSF) or milk of healthy people and of people with various diseases. Artificial catalytic Abs obtained by genetic engineering that were not found in humans were not included. Features of the formation of abzymes, depending on the conditions, are discussed in detail below.

#### 4.2.1. Autoimmune and Neurodegenerative Diseases

As can be seen from the data presented, AIDs are accompanied by the formation of abzymes with the broadest repertoire of catalytic activities. Catalytic Abs hydrolyzing DNA, RNA, oligonucleotides, proteins, peptides and oligosaccharides and possessing oxidoreductase activity are formed in these diseases [89,90,91,92,93,94,95,96,97,98,99,100,101,102,103,104,105,106,107,108,109,110,111,112,113,114]. It should be noted that the highest level of catalytic activity of abzymes is found in such severe autoimmune pathologies as systemic lupus erythematosus (SLE) and multiple sclerosis (MS) [89,90,91,92,93,94,95,96,97,98,99,100,101,102,103,104,105,106]. As is known, AIDs are characterized by spontaneous formation of Abs to a variety of autoantigens. For example, increased titers of anti-DNA Abs were detected in SLE in 38% of patients, and in MS in 17%–18% of patients, whereas DNA hydrolyzing abzymes were found in 90%–95% of patients with MS and SLE [20]. Thus, there is no correlation between the titer of autoAbs and the formation of catalytic Abs. The cause is an increase in the titers of specific high-affinity autoAbs, which occurs in the late stages of AID, or during exacerbation. Therefore, it is evident that the formation of Abs that specifically bind the antigen, and abzymes that bind and hydrolyze the antigen, occur in various clones of B-lymphocytes, as well as due to various mechanisms. Nevertheless, literature data are known to indicate that some sites of antigen hydrolysis are located in immunodominant regions of the molecule (i.e., to which Abs are formed). This suggests a common mechanism of the formation of abzymes and conventional Abs. For example, Abs that hydrolyze the basic myelin protein (MBP) were discovered MS, and five of the six main sites of MBP hydrolysis are in the immunodominant regions of the MBP molecule after Arg or Lys [101]. Thus, it can be assumed that with the development of AID, there is an increase in the number of clones of B-lymphocytes producing autoAbs, as well as abzymes.

B-lymphocytes are known to synthesize structurally and functionally different classes of Abs to varying stages of the life cycle. According to published data, IgMs are distinguished by their ability to catalyze the hydrolysis of model substrates at rates substantially higher than the IgG produced in the later stages of B cell differentiation after switching the synthesis of Ab isotypes [115]. For instance, IgM from the serum of patients with systemic lupus erythematosus (SLE) had 5–10 times higher DNA and RNA hydrolyzing activity than IgG from the same serum [91]. In another work, it was shown that the specific catalytic activity of IgM and sIgA from the sera of patients with MS in the MBP hydrolysis reaction is significantly higher than that of IgG [103]. S. Paul found that IgM has 344 times more proteolytic activity, and IgA has three times more activity than IgG [159,167]. Thus, according to the catalytic activity level, Igs of various classes can be arranged as follows: IgM > sIgA > IgG. It is expected that IgM has the highest activity, since this molecule has the least affinity for the substrate and a flexible antigen-binding domain. The high catalytic activity of IgM compared to IgG is not explained by the effects of avidity, since the multivalent binding of IgM prevents the low substrate concentrations used in the reaction. These data can be explained by the fact that proteolytic activity is lost during somatic hypermutation of the V domain after B cells switch from the generation of class M Abs to class G Abs [160]. Another explanation is the unique domain architecture of IgM, which helps maintain the integrity of the catalytic site. In the case of switching to the synthesis of another isotype, it leads to a decrease in catalytic activity [88].

Therefore, we can conclude that the formation of catalytic Abs is closely related to the presence of immunological disturbances and especially autoimmune processes. Available data indicate that the effective functioning of clones of B-lymphocytes producing abzymes starts in the presence of a pronounced autoimmune process [139]. Studies on mouse models confirm this. Thus, in MRL-lpr/lpr mice susceptible to spontaneous development of AID (SLE), the appearance of catalytic Abs was associated with profound impairments in the differentiation profile of bone marrow hematopoietic stem cells (BMHSC) and the suppression of cell apoptosis [168]. The highest increase in the level of abzyme activity was found in mice immunized with DNA. DNA hydrolyzing activity was shown to occur at the early stages of autoimmune pathology development when the level of autoAbs to DNA was still very low and comparable to that in healthy mice. A further increase in activity correlates with the development of clinical symptoms, as well as with an increase of anti-DNA Abs in serum [168]. Another study found that after immunization with a transitional hapten analog, MRL/lpr mice prone to SLE develop much more catalytic Abs than wild-type MRL/++, or BALB/c mice [85]. Thus, the formation of abzymes with DNase activity is an early and clear marker of autoimmune pathology.

Abzymes hydrolyzing specific proteins are formed with the autoimmune pathology progress. As indicated above, abzymes that hydrolyze MBP are formed in MS and SLE [93,95,101,102,103]. It was shown that abzymes hydrolyze histones H1, H2a, H2b, H3 and H4 in patients with MS [50], and histone H1 in patients with SLE [94]. Abzymes that break down thyroglobulin and the fluorescently labeled tripeptide Pro-Phe-Arg-MCA are revealed in Hashimoto’s thyroiditis [107]. The authors of this work note a low Km value for thyroglobulin (39 nM) in comparison with the tripeptide (18 μM) and suggest that the thyroglobulin binding step is specific and the hydrolytic reaction step is less precise. They also believe that the binding site and catalytic site are in immediate proximity. An interesting example of proteolytic abzymes is IgG and hydrolyzing factors VIII and IX of the blood coagulation system in patients with acquired hemophilia caused by spontaneous formation of autoAbs to these factors [111,112]. Moreover, in most cases, the hydrolysis of factor IX by Abs led to the activation of this factor, which can partially compensate for the Ab-mediated inhibition of endogenous factor VIII. Proteolytic abzymes are also formed in autoimmune myocarditis and hydrolyze cardiomyosin [114]. Another example of abzymes with proteolytic activity is IgM, hydrolyzing β-amyloid and peptide substrates in Alzheimer’s disease, which refers to neurodegenerative diseases [115]. Thus, abzymes with proteolytic activity can hydrolyze functionally important proteins and significantly affect the pathogenesis of autoimmune and neurodegenerative diseases.

#### 4.2.2. Inflammatory and Infectious Diseases

Catalytic Abs are also formed in inflammatory and infectious diseases. Generally, the study of natural catalytic Abs began with inflammatory conditions. Indeed, catalytic Abs were identified for the first time in patients with bronchial asthma in 1989 [116]. It was shown that autoAbs to the vasoactive intestinal peptide (VIP) hydrolyzed this antigen with an average catalytic efficiency (*k*_cat_ = 15.6 min^−1^) [116]. Another example is catalytic Abs that hydrolytically cleave factors VIII and IX of the blood coagulation system found in sepsis [117]. As the authors indicate, cumulative survival among patients with high IgG-mediated hydrolysis is higher than that of patients with low hydrolysis rates. An inverse correlation was also observed between the markers of severity of disseminated intravascular coagulation syndrome and the level of IgG activity of patients with sepsis. This indicates that catalytic IgG may be involved in the control of disseminated microvascular thrombosis leading to multiple organ failure. Another example of abzymes for infectious diseases is catalytic IgG and IgM of HIV patients hydrolyzing reverse transcriptase and HIV integrase [120,121,122,123,124], histones (H1, H2a, H2b, H3, H4) [125,126,127], β-casein [120] and different peptides. It was established that the detected proteolytic abzymes are a mixture of Abs subfractions that differ in biochemical parameters depending on the patient. Some abzymes possess serine protease-like catalytic activity, whereas others are metal-dependent, acidic or thiol proteases. Some of these Abs may exhibit protective properties, because they destroy viral proteins.

Abzymes that hydrolyze nucleic acids are also formed in viral and bacterial diseases. In particular, Abs hydrolyzing plasmid DNA were detected (pBluescript, pBR322) in HIV infection, hepatitis A, B, C and D, as well as tick-borne encephalitis [92,119,129]. RNA-hydrolyzing Abs that cleave cCMP, poly (U), poly (A), poly (C) and tRNA are also formed in hepatitis [92]. In addition to viral diseases, abzymes with nuclease activities were detected in bacterial infections. The level of DNase activity decreased in the following order: purulent surgical infections caused by *Staphylococcus epidermidis* > purulent surgical infections caused by *Staphylococcus aureus* > shigellosis > meningococcal meningitis > urogenital chlamydia associated with arthritis (Reiter’s disease) > streptococcal infection [130]. DNase activity was absent in Abs from healthy donors, as well as patients with influenza, pneumonia, tonsillitis, tuberculosis, duodenal ulcer and some types of cancer (tumors of the uterus, breast, stomach and intestines) [92]. Importantly, the level of catalytic activity of Abs in inflammatory and infectious diseases is lower than in AIDs [92].

#### 4.2.3. Cancer

Catalytic Abs are found only in tumors of the lymphatic system associated with the proliferation of B cells: multiple myeloma, chronic lymphocytic leukemia, mantle cell lymphoma, marginal zone lymphoma and follicular lymphoma [135]. There was no DNA hydrolyzing activity in diseases associated with the proliferation of T-lymphocytes (with non-Hodgkin’s T-cell lymphoma, Cesari’s syndrome, acute T-lymphoblastic leukemia and Hodgkin’s lymphoma) [135]. These data indicate an increase in the likelihood of clones producing DNA-hydrolyzing abzymes in tumors derived from relatively mature B cells compared to other types of malignant lymphoproliferation [135]. Furthermore, it was found that DNA abzymes are predominantly present in the blood of patients with lymphoproliferative diseases complicated by various autoimmune disorders [135]. This observation suggests similar mechanisms for the formation of abzymes in tumors and AIDs. In addition to DNA-hydrolyzing catalytic Abs, abzymes with the proteolytic activity, which hydrolyze prothrombin [134] and β-amyloid [115] are formed in multiple myeloma and Waldenstrom macroglobulinemia [133]. Desialylating Abs are an interesting example of abzymes generated in multiple myeloma [132]. Modification of the level of surface sialylation molecules by these Abs can lead to a change in tumorigenicity and metastatic potential of cells, as well as facilitating the removal of apoptotic cells.

#### 4.2.4. Alloimmune Diseases

Catalytic Abs with proteolytic activity were detected in the case of alloimmune diseases. For example, Abs that hydrolyze factor VIII of the coagulation system were found in hemophilia A after replacement therapy [136,137]. Abzymes that hydrolyze coagulation factors VIII and IX, but not factor VII and prothrombin, are produced in patients with a kidney rejection reaction during transplantation. This indicates the participation of catalytic Abs in the regulation of the coagulation state [138].

#### 4.2.5. Metabolic Diseases

Abzymes were found only in diabetes mellitus. In particular, Abs hydrolyzing DNA [139] and β-casein [140] were found in patients with diabetes. Abs and immune complexes are formed against glucose-regulated protein 94 (Grp94), which refers to heat shock proteins in patients with type I diabetes [140]. In addition, the formation of Fab/(Fab)_2_ adducts with Grp94 was detected, as well as anti-idiotypic Abs against these adducts, which inhibited proteolytic activity [140]. Thus, this is the first example of anti-idiotypic Abs, which inhibit the proteolytic activity of Abs.

#### 4.2.6. Psychiatric Disorders

Some mental disorders are inextricably linked to chronic low-grade inflammation and immune system dysfunction [169]. Additional evidence of immune disturbances in mental disorders is the formation of catalytic Abs. For example, autism spectrum disorder shows the presence of IgG and IgA hydrolyzing MBP [141]. Perfusion of these abzymes to rats induces a decrease in long-term potentiation in the hippocampus and impairs synaptic plasticity [141]. We also found evidence of humoral immune system dysfunction in schizophrenia. A light chain of Igs with reduced molecular weight (17.3 kDa) in some patients with schizophrenia, similarly to patients with SLE, was identified by MALDI-TOF mass spectrometry [142]. Increased titers of Abs to DNA and MBP are found in patients, compared to healthy donors [142,146]. Additionally, catalytic Abs capable of hydrolyzing DNA [142,143], RNA and microRNA [143,144,145] are found in patients with schizophrenia. These data indicate the formation of antinuclear Abs. They are consistent with the available data [170]. The level of catalytic activity of Abs is correlated with the clinical parameters of schizophrenia [142,145,146]. Furthermore, the formation of Abs hydrolyzing MBP is indicated in schizophrenia [146]. Such catalytic anti-MBP Abs can be associated with impaired myelination in schizophrenia. In addition, IgG with redox activity, which may be involved in protecting against oxidative stress, was detected [147]. However, the role of catalytic Abs in psychiatric disorders is still not fully understood.

#### 4.2.7. Normal Physiological Conditions

For a long time, it was believed that abzymes do not form in healthy people. The first example of abzymes in the absence of any disease was sIgA of human milk, catalyzing protein phosphorylation [59,154]. Subsequently, lipid kinase [155] and polysaccharide kinase [156] activities of human IgG and sIgA were detected. Thus, natural abzymes that catalyze not a degradation reaction but a synthesis reaction were identified for the first time. Moreover, these reactions are bisubstrate reactions. Interestingly, Abs can use ATP and other nucleotides (dATP, GTP, dGTP, UTP, TTP), including orthophosphate, as a donor of the phosphate group.

IgG and sIgA hydrolyzing DNA [59,148,151], RNA [148,151], oligonucleotides [148,150,151] and microRNA [157,158] were found in human milk. The presence of abzymes hydrolyzing β-casein [152] and polysaccharides [96,149] are also shown in the milk of healthy women. It should be noted that human milk Abs are characterized by higher activity compared to most known abzymes from the serum of patients with AIDs. It is assumed that catalytic Abs hydrolyzing nucleic acids and polysaccharides can destroy the DNA and RNA of viruses, as well as the elements of the capsule of bacteria, thereby participating in protecting the newborns from viral and bacterial infections. In a recent study it was shown that microbiome-derived single-stranded fecal RNA is a natural ligand for cation channel Piezo1, which regulates systemic serotonin production in gastrointestinal enterochromaffin cells [171]. Colonic infusion of RNase A decreased gut motility but increased bone mass; thus, Piezo1 ultimately regulates the gut peristalsis and bone homeostasis [171]. Considering our data that human milk sIgA and IgG hydrolyze RNA and microRNA [148,150,151,157,158], it can be assumed that these immunoglobulins are involved in the regulation of peristalsis and bone metabolism in newborns through the hydrolysis of fecal single-stranded RNA. In general, human milk abzymes may play an important role in protecting the newborns from different pathogens and gut homeostasis regulation.

Proteolytic abzymes were found in the blood serum of clinically healthy donors. For example, catalytic IgG and IgM hydrolyzing the HIV envelope glycoprotein gp120 have been shown in uninfected people [160,161,162]. One of the cleavage sites (Lys432-Ala433) is located in the superantigenic determinant. Furthermore, IgM from healthy donors can hydrolyze the transthyretin protein, which forms aggregates and leads to amyloidosis development [163]. Abzymes that hydrolyze the extracellular fibrinogen binding protein of *Staphylococcus aureus* were also discovered [164]. Thus, catalytic Abs of healthy donors can help protect the organism and be the first line of defense against various pathogens.

Abs of healthy donors also possess catalase, peroxidase and oxidoreductase activities and oxidize various toxic compounds [147,165,166]. The redox activity of such Abs is due to the presence of bound transition metal ions and oxidizable thiol groups in the IgG molecules. Along with the well-known antioxidant enzymes, these redox active catalytic Abs may be involved in redox-state correction and protection from oxidative stress.

Consequently, we can conclude that the formation of catalytic Abs is an integral property of the immune system. However, an increase in the spectrum of the generated abzymes, (especially nucleic acids and specific proteins hydrolyzing Abs), occurs with profound disturbances in the immune system. Autoimmune reactions, inflammatory and infectious processes and the malignant transformation of B cells lead to an increase in the level of catalytic Abs. However, the highest increase in the catalytic activity of Abs occurs with autoimmune pathology. The presence of catalytic Abs that hydrolyze specific proteins and nucleic acids is the earliest sign of immunological disorders.

## 5. The Biological Role of Catalytic Abs and Their Use in Medicine and Biotechnology

Based on the data presented in Section 4.2, the formation of catalytic Abs is a common immunological phenomenon. In this regard, the question arises of the functional role of catalytic Abs in the immune system. On the one hand, they can play a beneficial role; on the other hand, a pathological one. The biological function of abzymes depends on the specific disease and the mechanism of their formation.

The pathological role of abzymes hydrolyzing MBP is evident in MS. It was shown that the rates of MBP catalysis by autoAbs are among the highest recorded for abzymes and, therefore, are sufficient for their pathological effects with the slow development of neurodegeneration during the progression of MS [101]. The proteolysis of thyroglobulin by abzymes can lead to dysfunction of the thyroid gland in Hashimoto’s thyroiditis [107]. Hydrolysis of factor VIII of the blood coagulation system by autoAbs of patients with hemophilia A during replacement therapy disrupts the coagulation cascade and contributes to the development of bleeding [136,137]. However, proteolysis of prothrombin under the action of light chains of Abs in patients with multiple myeloma leads to the formation of fragments that activate the conversion of fibrinogen to fibrin, which contributes to thrombosis [134]. A decrease in the concentration of a vasoactive intestinal peptide due to hydrolysis by proteolytic abzymes can lead to impaired relaxation of the smooth muscles of the bronchi and respiratory dysfunction in bronchial asthma [116]. The pathological effects of DNA hydrolyzing abzymes are due to complement-independent cytotoxicity. It has been shown that DNA hydrolyzing Abs are able to cross the plasma and nuclear membranes of various cells, cause DNA fragmentation and activate cell apoptosis through caspase-dependent mechanisms [172,173,174]. Not only fragments of Igs (light chains, Fab domains) [175], but also whole Ab molecules [173,176] are capable of translocation through the membrane. The penetration of Igs into the cell is due to interaction with sulfated proteoglycans on the cell surface by caveolae-mediated endocytosis [174]. An alternative implementation of the cytotoxic activity of DNA abzymes is their interaction with receptors on the cell surface that cause apoptosis [177]. However, not all DNA hydrolyzing abzymes are cytotoxic. It was shown that DNA abzymes obtained by immunization of healthy animals or from patients with various bacterial and viral infections are not cytotoxic to the tumor and normal cells [92].

However, along with this, there are many examples of the positive role of abzymes. Hydrolytic cleavage of β-amyloid under the action of proteolytic IgM reduces the aggregation and toxicity of this protein in neuronal cell cultures, thereby preventing the development of Alzheimer’s disease [115]. As described in Section 4.2.2, Abs produced during sepsis, hydrolyzing coagulation factors VIII and IX, can be involved in the regulation of disseminated intravascular coagulation [117]. Degradation of thyroglobulin from the bloodstream by abzymes helps to minimize the autoimmune response and reduce harmful immunocomplex effects in Hashimoto’s thyroiditis [107]. Abzymes hydrolyzing integrase and HIV reverse transcriptase also perform a protective function [120,124]. IgM hydrolyzing transthyretin with aberrant conformation, dissolving toxic species of this protein and protecting against the development of amyloidosis, has been found in healthy donors [163]. Serum Abs from healthy, uninfected donors have the ability to hydrolytically cleave HIV gp120 protein [160,161] and extracellular fibrinogen binding protein *S. aureus* [164]. The authors of the article [162] believe that since mucosal IgA and serum IgM have high catalytic activity and neutralize HIV, they constitute the first line of defense against HIV and other pathogens. Abzymes with DNA and RNA hydrolyzing activity are formed in pregnancy, which help the immature immune system of the newborn to defend itself against infections by destroying the nucleic acids of viruses and bacteria [176]. Another example of the protective function of human milk proteolytic abzymes is their ability to activate the synthesis of defensins in intestinal epithelial cells by hydrolysis of a protease-activated receptor 2 [153]. Abzymes with oxidoreductase activity may be involved in the regulation of oxidative stress [165,166]. These data indicate the potential protective function of constitutively produced catalytic Abs. This kind of “catalytic immunity” partially encoded in the germline genes provided evolutionary advantages in survival [163].

In general, abzymes with the same disease can play a dual role. This is well illustrated by the example of DNA hydrolyzing abzymes, which on the one hand can be cytotoxic, and, on the other hand, protect against viral infections. Therefore, it is especially important to study the entire spectrum of catalytic activities of Abs in a particular disease to find out their role in this pathology.

The biopharmaceutical industry has made significant strides in the development of Ab-based biotherapeutics during 2010–2020 [178]. The unique spatial structure of catalytic Abs, combining high specificity and functional activity, opens up great opportunities for the development of therapeutic agents based on them. Despite the low rate of catalysis compared to classical enzymes and the inability to implement the complex dynamic enzyme-specific mechanisms of catalysis, abzymes have many advantages. First, catalytic Abs have low immunogenicity. Secondly, Igs circulate in the blood for a long time (more than two weeks), in contrast to canonical enzymes, which, when released into the bloodstream, are rapidly inactivated by proteases [179]. Furthermore, Abs are able to cross the histohematological barrier and accumulate in the affected organs, in places of inflammation. Thirdly, Igs are in high concentration compared to enzymes in the blood. Fourth, abzymes have high specificity, which can be enhanced by various genetic engineering methods. Fifth, abzymes can bind and destroy many antigen molecules due to their avidity, whereas classical Abs can bind only a few antigens. This allows clinicians to reduce the number of injected Ab molecules to provide significant effects, which, in turn, reduces the risk of side effects and the cost of therapy. The rapid removal of antigens from the bloodstream by abzymes also helps to minimize autoimmune reactions and reduce the pathological immunocomplex effects of classical Abs. Sixth, abzymes can be obtained in large quantities using biotechnological methods. Seventh, thanks to genetic engineering methods, it is theoretically possible to create abzymes for almost any antigen.

There are several reviews of the use of abzymes in medicine and biotechnology [60,86,177,180,181,182]. Certain hopes are associated with the prospect of using abzymes to treat addiction to psychoactive substances. In particular, monoclonal abzymes that hydrolyze cocaine and nicotine are being developed [60,180]. Abzymes that selectively hydrolyze organophosphorus compounds are beimg created. Thus, a catalytic Ab A17 that hydrolyzes organophosphorus pesticide paraoxon was developed using combinatorial chemistry methods, ultrahigh-throughput screening techniques and quantum mechanical calculations (QM/MM calculations) for the maturation of Abs in silico [183,184]. Abzymes can also be used to protect against various infections: an HIV vaccine is being developed based on the ability of abzymes to hydrolyze the gp120 viral protein [182]. The HIV gp41 sheath glycoprotein is another target for the development of catalytic Abs. Moreover, a catalytic Ab that hydrolyzes the polysaccharides of the capsule of the bacterium *Cryptococcus neoformans* is being developed [185]. It was found that the obtained monoclonal Ab 18B7 also effectively cleaves the polysaccharides of fungal cells. A monoclonal Ab UA15 that hydrolyzes urease of the gastritis-causing *Helicobacter pylori* bacteria has been developed [86]. The use of this Ab has been shown to significantly reduce the amount of this bacterium in the stomach of mice. Monoclonal light chains were obtained from a person vaccinated against rabies virus that possessed protease activity and protected against infection with this virus [60]. Developed DNA hydrolyzing abzymes 22F6-L are capable of protecting against type A influenza virus (H1N1) [86]. Abs that hydrolyze β-amyloid are produced in Alzheimer’s disease. A new method has been developed for the delivery of an adeno-associated viral vector encoding the gene for β-amyloid-specific catalytic Ab rAAV9-IgVL5D3 [186]. Targeted Ab gene delivery and expression directly in the brain is a safer and more effective approach to the prevention and treatment of Alzheimer’s disease than conventional anti-β-amyloid Abs. Monoclonal Ab ETNF-6-H hydrolyzing tumor necrosis factor α has been developed for anti-cytokine therapy of AIDs [180]. A Se-containing catalytic single-chain variable fragment (scFv) named Se-scFv-2D8 with glutathione peroxidase activity was designed to correct oxidative stress [180]. Lee, with colleagues, created the single-domain catalytic Ab 3D8 VL hydrolyzing nucleic acids [187]. Ab 3D8 VL significantly reduced the mRNA level of the target Her2 gene in breast carcinoma cells. Thus, a sequence-specific, nucleotide-hydrolyzing, cell-penetrating Ab has been created for specific gene silencing.

In addition to the above, abzyme prodrug therapy is an exciting option for the use of catalytic Abs [60,180]. The basis of this method is the technology of Ab-directed enzyme prodrug therapy. The method is based on the use of Abs against antigens of tumor cells conjugated with enzymes to directly activate the prodrug form of the drug near the tumor cells. Replacing an enzyme with an abzyme reduces immunogenicity. The most studied abzyme for prodrug therapy is the Ab 38C2, which has aldolase activity and activates the prodrug form of the antitumor drugs doxorubicin and camptothecin. Ab 38C2 has been shown to inhibit the growth of human colon carcinoma and prostate cancer lines and reduce tumor growth in an animal model of neuroblastoma [60]. In addition, 38C2 can be used in diabetes mellitus to activate aldol-modified insulin. At least four monoclonal Abs conjugates (CVX-045, CVX-060, CVX-096 and CVX-241) based on humanized 38C2 and obtained by chemical programming strategies were used in clinical trials [188]. It was shown that CVX-045 (mimics thrombospondin-1), CVX-060 (neutralizes angiopoietin-2), CVX-096 (mimics glucagon-like peptide-1) and CVX-241 (as bispecific Abs neutralizes angiopoietin-2 and vascular endothelial growth factor) were generally well tolerated [188]. CVX-060 was also tested in combination with multi-targeting inhibitors of receptor tyrosine kinases in multicenter phase I/II clinical trials (ClinicalTrials.gov: NCT00982657 and NCT01441414) for the treatment of renal cell carcinoma. CVX-096 was evaluated for the treatment of type II diabetes mellitus in phase I clinical trials (NCT00886821). However, to date, none of these Abs have passed into other phases of clinical trials. Additionally, an HIV catalytic vaccine that induces the induction of abzymes in patients with acquired immune deficiency syndrome is being developed with funding from the Abzyme Research Foundation. Clinical trials are scheduled for 2021.

The pandemic spread of novel coronavirus pneumonia (COVID-19) has heightened the global need to develop effective treatments. The special properties of catalytic Abs can be used to develop a catalytic vaccine for COVID-19 similar to the HIV catalytic vaccine (see earlier). Furthermore, some patients with severe COVID-19 are known to exhibit significantly increased levels of pro-inflammatory cytokines (“cytokine storm” syndrome or cytokine release syndrome) [189]. Cytokine release syndrome leads to massive neutrophil infiltration of the lungs, diffuse damage of the alveolar wall and ultimately to respiratory or multiple organ failure [189]. Therefore, monoclonal catalytic Abs can be developed for anti-cytokine therapy. However, patient stratification is needed to identify the subgroup of patients in whom anti-cytokine therapy can improve disease outcome. Retargeting existing approved anti-cytokine therapeutic agents may be a more rapid solution to this problem. Nonetheless, anti-cytokine catalytic Abs potentially have many therapeutic applications.

Thus, since Abs represent the largest class of biological therapy drugs on the pharmaceutical market [178], it is likely that abzymes will become a source of new therapeutic tools in medicine and biotechnology.

## 6. IVIg-Mediated Effector Functions Are Determined by Canonical and Non-Canonical Functions of Abs

Intravenous immunoglobulin (IVIg) is a pooled preparation of normal Igs obtained from thousands of healthy donors [190]. In addition to IgG monomers (> 96%), IVIg preparations contain a small percentage of IgG, IgM and IgA dimers [5]. Initially, IVIg preparations were used for substitution therapy in immunodeficiency conditions accompanied by a deficiency of Abs, but now their scope is expanding [190,191]. Indications for IVIg therapy include autoimmune pathology such as Kawasaki disease, thrombocytopenic purpura, chronic inflammatory demyelinating polyneuropathy, multifocal motor neuropathy, etc. [5]. However, many clinical trials have shown the effectiveness of IVIg in the treatment of other autoimmune and inflammatory diseases, including systemic vasculitis, myasthenia gravis, autoimmune hemolytic anemia, dermatomyositis, discoid lupus erythematosus, graft-versus-host disease, etc. [5,192,193]. Prospects for the use of IVIg preparations are associated with their complex effector functions, including effects on soluble mediators, as well as on the cellular components of the immune system (extensively reviewed in [5,190,193]). The effector functions of IVIg are due to the effects of both Fab and Fc domains of Igs. Fab-mediated effects include inhibition of autoAbs by idiotypic–antiidiotypic networks and neutralization of the activated components of the complement system [5]. Fc-mediated effector functions are due to interactions with FcRs and other receptors [5]. Thus, many of the effector functions of IVIg are due to the canonical functions of the Abs described previously (see Table 1). However, non-canonical functions of Abs also contribute to the realization of the IVIg effector functions. For instance, many effects of IVIg preparations on granulocytes may be due to receptor agonist activity [194]. In particular, Abs to tumor necrosis factor receptors contained in IVIg preparations can lead to apoptotic death of granulocytes [194]. We hypothesize that catalytic Abs present in the serum of some healthy donors may also contribute to the realization of the effects of IVIg preparations. For example, catalytic Abs hydrolyzing extracellular fibrinogen binding protein *S. aureus*, which are constitutively present in healthy non-immunized donors, will help protect against various infections [164]. Moreover, we suggest that IVIg preparations can be additionally enriched with various monoclonal Abs with both canonical and non-canonical functions. As indicated previously, many monoclonal catalytic Abs are effective. Therefore, enrichment of IVIg preparations with monoclonal Abs can have attractive prospects.

## 7. Conclusions

The use of next-generation sequencing technologies has revealed an extraordinary Ab repertoire diversity in humans. This vast repertoire of Igs is reflected in a wide variety of Ab types: antigen-specific, natural, polyreactive, broadly neutralizing, homophilic, bispecific, catalytic, etc. Catalytic Abs are an important component of this repertoire. Such a variety of Ab types produced by the B cells allows them to realize both canonical and non-canonical biological functions. Knowledge about the variety of functions performed by Abs has broad translational prospects. In particular, the therapeutic effects of IVIg are due to both canonical and non-canonical Ab functions. The special properties of some specific Abs (e.g., catalytic) can be used to create new therapeutic tools. The presence of catalytic Abs is often associated with autoimmune disorders. Therefore, data on catalytic Abs can be used for diagnostic purposes.

## Figures and Tables

**Figure 1 ijms-21-05392-f001:**
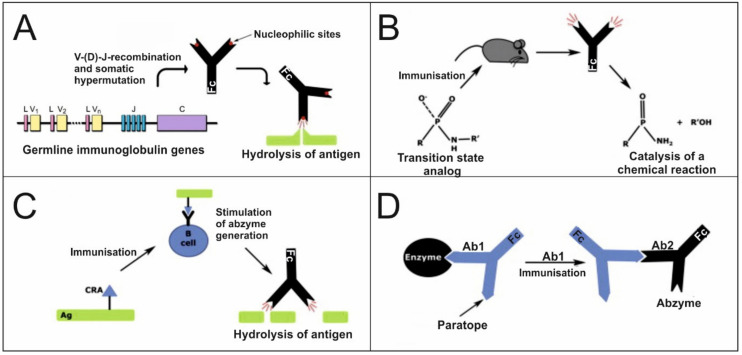
Various mechanisms of the formation of catalytic Abs. (**A**) Adaptive maturation of nucleophilic centers of Abs, originating from the initial germline Ig genes; (**B**) Immunization with a stable analog of the transition state of a chemical reaction; (**C**) Immunization with an antigen conjugated to a covalent reactive analog (CRA); (**D**) The formation of abzymes due to the idiotypic–anti-idiotypic interactions of Abs. Adapted with permission from Elsevier [60].

**Table 1 ijms-21-05392-t001:** The canonical and non-canonical functions of antibodies (Abs).

Ig Classes	Canonical Functions of Abs ^1^	
	Fab-Dependent	Fc- and Whole Ab-Dependent
IgG-mediated	Agglutination, neutralization, and excretion of specific antigens Similarly sIgA and sIgM, IgG is involved in controlling the diversity of certain commensal and pathogenic microorganisms	Ab-mediated complement activation Ab-dependent cell-mediated cytotoxicity Ab-dependent cellular phagocytosis through the interaction of IgG with FcγRs, DC-SIGN on macrophages Participation in antigen processing as a result of FcγRs-mediated internalization of immune complexes Intracellular Ab-mediated degradation of antigen in the proteasome after interaction with C1q and TRIM21 Ab-mediated immunomodulation, including the release of pro-inflammatory molecules, activation, differentiation and development of immune cells after interaction of IgG with FcγRs B cell selection and survival, regulation of plasma cell apoptosis, control of IgG production through the interaction of IgG with FcγRs and CD23 Provide specific protection for newborns from certain pathogens as a result of transport of mother’s IgG through FcRn
IgM-mediated	Agglutination, neutralization, and excretion of various pathogens sIgM, together with sIgA regulate bacterial intestinal diversity	Pronounced Ab-mediated complement activation Together with complement, provides transport of antigens to secondary lymphoid organs and the initiation of an immune response Strengthening phagocytosis and promoting the presentation of antigens Stimulation of macrophage uptake of apoptotic cells and their degradation products IgM through FcμR is involved in the regulation of B cell development and IgG production, as well as immune tolerance
IgA-mediated	sIgA promotes the opsonization, agglutination, and excretion of pathogenic microorganisms and their products at mucosal surfaces (immune exclusion) Low-affinity sIgA retain commensal bacteria in the intestinal lumen (immune inclusion) sIgA neutralizes intracellular pathogen determinants in the epithelial-cell endosomes	Serum monomeric IgA promotes anti-inflammatory response after interaction with FcαRI and other receptors IgA immune complexes contribute to a pro-inflammatory response after interaction with FcαRI and PRR Serum monomeric IgA, but not sIgA, initiate bacterial phagocytosis due to interaction with FcαRI Antigens excretion due to the secretion of sIgA through the interaction of the sIgA-antigen complex with pIgR and its release into the lumen of the mucosa Colostrum and milk IgA provide neonatal intestinal homeostasis
IgE-mediated	IgE opsonize and mediate the destruction and removal of helminths and other pathogens IgE contribute to the inactivation of animal poisons and toxins	IgE initiates mast cell and basophil degranulation, as well as the synthesis of inflammatory mediators and the secretion of cytokines and chemokines after antigen recognition by IgE, associated with FcεRI Regulate growth, maturation and survival and mast cell homeostasis even in the absence of antigen Regulate the expression of FcεRI and CD23 Participate in the transport of allergen from the intestine to the mucous membrane due to the interaction of the allergen-IgE complex with CD23 or FcεRI, thereby facilitating the presentation of the antigen Stimulate Th2 response and suppression of Treg generation
IgD-mediated	Secreted IgD is involved in the regulation of commensal and pathogenic bacteria and mucosal allergens	IgD associated with basophils and other cells, after antigenic stimulation, triggers the release of IL-4, which causes the production of IgG by B cells IgD receptors expressed on B cells regulate their development and maturation, as well as clonal anergy and self-tolerance
**Non-Canonical Functions of Abs ^1^**		
**Ig Classes**	**Fab-Dependent**	**Fc- and Whole Ab-Dependent**
IgG-mediated	Antigen cleavage due to the catalytic activity of IgG Direct inactivation of pathogens in the absence of effector cells and molecules Cofactor effects of IgG in neutralizing pathogens Triggering of cell signaling through receptor agonist activity of Igs Compensation of innate immune defects due to anti-cytokine activity or other mechanisms Carriage, bioavailability regulation and protection of hormones from proteolytic degradation ROS detoxification due to redox activity of IgG	Modulation of intracellular insulin signaling due to the interaction of the hyposialylated IgG Fc domain with FcγRIIb Ab-dependent enhancement of infection or disease
IgA-mediated	Antigen cleavage due to the catalytic activity of IgA Intestinal sIgA regulates the penetration of microbial metabolites into the systemic circulation involved in regulating the metabolism and immunity of the host High avidity pathogen-specific sIgA contribute to the formation of bacterial clusters, “enchained growth” and enhanced clearance	Transepithelial transfer of bacteria from the small intestine to Peyer’s patches and induction of T cell-dependent Ab responses The Fc domain of sIgA, interacting with bacterial glycans, modulates the expression of polysaccharide utilization loci, including MAFF
IgM-mediated	Antigen cleavage due to the catalytic activity of IgM Direct inactivation of pathogens in the absence of effector cells and molecules	Modulation of lymphocyte intracellular signaling due to the interaction of the Fc domain of IgM with FcμR

1 The table was compiled based on the works [1,11,12,13,14,15,16,17,18,19,20,21,22,23,24,25,26,27,28]. Abbreviations: C1q—complement component 1q, CD—a cluster of differentiation, DC-SIGN—dendritic cell-specific intercellular adhesion molecule 3 grabbing non-integrin, Ig—immunoglobin, IL—interleukin, FcRn—neonatal Fc receptor, FcαRI—Fc receptor for IgA, FcγRs—Fc receptors for IgG, FcεRI—Fc receptor for IgE, FcμR—Fc receptor for IgM, MAFF—mucus-associated functional factor, pIgR—polymeric immunoglobulin receptor, PRR—pattern recognition receptor, ROS—reactive oxygen species, sIgA—secretory immunoglobulin A, TRIM21—tripartite motif-containing protein 21, Th2—type 2 helper T cells, Treg—regulatory T cells.

**Table 2 ijms-21-05392-t002:** Comparison of the origin and features of various Ab types in humans.

Abs Type	Origin	Affinity	Specificity	Biological Roles
Antigen-specific adaptive Abs	B2 cells	High	High	The binding of a specific pathogen Ab-mediated complement activation Ab-dependent cell-mediated cytotoxicity Ab-dependent phagocytosis Regulation of immune cells homeostasis
Natural Abs	B1 cells and marginal zone B cells	Low	Low	Direct pathogen neutralization Classical complement activation Antigen transport to secondary lymphoid organs and presentation Ab-dependent cell-mediated cytotoxicity Phagocytosis of apoptotic cells Clearance of DAMPs and prevention of autoimmunity Regulation of immune cells homeostasis
Polyreactive Abs	B1 cells	Low	Moderate	The same functions as natural Abs
Broadly neutralizing Abs	B1 cells	Low	Moderate	The same functions as natural Abs
Homophilic Abs	B2 cells	High	High	The same functions as antigen-specific Abs
Bispecific Abs	B2 cells	High	High	The same functions as antigen-specific Abs
Catalytic Abs	Unknown, presumably B1 cells	Low	Moderate	Hydrolysis of antigen ROS detoxification due to redox activity Promoting of autoimmune reactions Minimization of inflammatory reactions

**Table 3 ijms-21-05392-t003:** Comparison of the origin and features of various Ab types in humans.

Disease/Condition	Catalytic Activity of Abs	Substrate	References
Autoimmune and Neurodegenerative Diseases			
Systemic lupus erythematosus	DNA-hydrolyzing	DNA plasmid pUC19 *, d(pA)_10_ *, d(pA)_13_	[89,90,91,92,93,94,95,96,97]
	RNA-hydrolyzing	p(A)_13_, p(U)_10_, poly(А), poly(C), poly(U), сСМР, yeast RNA	
	Proteolytic	MBP **, OP_17_-MBP, OP_19_-MBP, histone H1**	
	Amololytic	Different maltooligosaccharides	
	Peroxidase and oxydoreductase	DAB, ATBS, OPD, pHQ and others in the presence and absence of hydrogen peroxide	
Multiple sclerosis(IgG, IgA and IgM from blood serum and CSF)	DNA-hydrolyzing	HeteroODN_15_, d(pT)_10_	[96,97,98,99,100,101,102,103,104,105,106]
	RNA-hydrolyzing	Poly(А), poly(C), poly(U), сСМР	
	Proteolytic	MBP, OP_85-101_-MBP, H-Pro-Phe-Arg-MCA, histones H1, H2а, H2b, H3, H4	
	Amilolytic	Different maltooligosaccharides	
	Peroxidase and oxydoreductase	DAB, ATBS, OPD, pHQ and others in the presence and absence of hydrogen peroxide	
Hashimoto’s thyroiditis	Proteolytic	Pro-Phe-Arg-MCA, thyroglobulin	[107,108,109]
	DNA-hydrolyzing	DNA plasmid pBR322	
	RNA-hydrolyzing	poly(А), poly(C), poly(U), сСМР, yeast RNA, тРНК^Phe^, тРНК^Lys^	
Rheumatoid arthritis	Proteolytic	Pro-Phe-Arg-MCA and other MCA-labeled peptides	[110]
Systemic scleroderma	DNA-hydrolyzing	DNA plasmid pUC19	[89]
Acquired hemophilia	Proteolytic	Factor-VIII, factor-IX	[111,112]
Spondyloarthropathy, polyarthritis	DNA-hydrolyzing	Calf thymus DNA, DNA plasmid pBR322	[109,113]
	RNA-hydrolyzing	Poly(А), poly(C), poly(U), сСМР, yeast RNA, тРНК^Phe^, тРНК^Lys^	
Autoimmune myocarditis	Proteolytic	Cardiomyosin	[114]
	DNA-hydrolyzing	Plasmid DNA	
Alzheimer’s disease	Proteolytic	β-amyloid, Glu-Ala-Arg- MCA	[115]
**Inflammatory and Infectious Diseases**			
Bronchial asthma	Proteolytic	VIP	[116]
Sepsis	Proteolytic	Factor-VIII, factor-IX, Pro-Phe-Arg-MCA, HMGB1 protein	[117,118]
HIV-infection (IgG and IgM)	DNA-hydrolyzing	DNA plasmid pBluescript	[119,120,121,122,123,124,125,126,127,128]
	Proteolytic	β-casein, reverse transcriptase and integrase, HIV **, histones ** H1, H2а, H2b, H3, H4, different peptides	
Hepatitis А, В, С, D	DNA-hydrolyzing	DNA plasmid pBR322	[92]
	RNA-hydrolyzing	cCMP, poly(U), poly(A), poly(C), тРНК^Phe^	
Tick-borne encephalitis	DNA-hydrolyzing	DNA plasmid pBluescript	[129]
Streptococcal infection	DNA-hydrolyzing	DNA plasmid pBluescript	[130]
Urogenital Chlamydia Associated with Arthritis	DNA-hydrolyzing	DNA plasmid pBluescript	[130]
Meningococcal meningitis	DNA-hydrolyzing	DNA plasmid pBluescript	[130]
Shigellosis	DNA-hydrolyzing	DNA plasmid pBluescript	[130]
Purulent surgical infections caused by *Staphylococcus aureus* and *Staphylococcus epidermidis*	DNA-hydrolyzing	DNA plasmid pBluescript	[130]
Genitourinary ureaplasmosis associated with reactive arthritis	DNA-hydrolyzing	DNA plasmid pBluescript	[130]
Influenza (light chains)	Nuclease	DNA plasmid pBR322, genome RNA from Noda virus	[131]
	Proteolytic	Peptide-AMC	
**Cancer**			
Multiple myeloma (light chains)	Sialidase	2′- (4-methylumbelliferyl) -α-d-*N*-acetylneuraminic acid	[132,133,134,135]
	Proteolytic	BApNA, prothrombin	
	DNA-hydrolyzing	DNA plasmid pUC19	
Chronic lymphocytic leukemia	DNA-hydrolyzing	DNA plasmid pUC19	[135]
Mantle cell lymphoma	DNA-hydrolyzing	DNA plasmid pUC19	[135]
Marginal area lymphoma	DNA-hydrolyzing	DNA plasmid pUC19	[135]
Follicular lymphoma	DNA-hydrolyzing	DNA plasmid pUC19	[135]
Waldenstrom macroglobulinemia	Proteolytic	β-amyloid	[135]
**Alloimmune Diseases**			
Hemophilia A (after replacement therapy)	Proteolytic	Factor-VIII	[136,137]
Transplant rejection reaction	Proteolytic	Pro-Phe-Arg-MCA, factor-VIII, factor-IX	[138]
**Metabolic Diseases**			
Diabetes	DNA-hydrolyzing	DNA plasmid pBluescript	[139,140]
	Proteolytic	BApNA, β-casein	
**Psychiatric Disorders**			
Autism (IgA, IgG, and IgM)	Proteolytic	MBP**, D-Ile-Pro-Arg-pNA, D-Leu-pNA, other	[141]
Schizophrenia	DNA-hydrolyzing	DNA plasmid pBluescript	[142,143,144,145,146,147]
	RNA-hydrolyzing	cСMP, poly(С), poly(А), yeast RNA, microRNA: miR-137, miR-9-5p, miR-219-2-3p, miR-219a-5p	
	Proteolytic	MBP, different peptides	
	Catalase-like	Hydrogen peroxide	
**Normal Physiological Conditions**			
Pregnancy and feeding a newborn (sIgA and IgG from milk and blood serum)	DNA-hydrolyzing	DNA plasmid pBR322, Phage λ DNA, тРНК^Lys^, d(pA)_10_, d(pT)_10_, d(pC)_10_	[59,92,148,149,150,151,152,153,154,155,156,157,158]
	RNA-hydrolyzing	r(A)_10_, r(T)_10_, r(C)_10,_ microRNA: miR-137, miR-219a-5p, miR-219-2-3p, and miR-9-5p	
	Amilolytic	Different maltooligosaccharides	
	Nucleotide hydrolyzing	ATP, GTP, CTP, dATP, dGTP, dCTP, AMP, etc.	
	Proteolytic	β-casein, BSA, activated protease receptor 2 *, BApNA	
	Proteinkinase	β-casein in the presence of γ-[^32^Р]NTP or γ-[^32^Р]dNTP	
	Lipid kinase	lipids in the presence of γ-[^32^Р]ATP and γ-[^32^Р]Pi	
	Oligo- and polysaccharide kinase	Oligo- and polysaccharides in the presence of γ-[^32^Р]ATP and γ-[^32^Р]Pi	
Healthy condition (IgA, IgG, and IgM)	Proteolytic	Pro-Phe-Arg-MCA, Glu-Ala-Arg-AMC, etc.; HIV gp120 protein, transthyretin, extracellular fibrinogen binding protein *S. aureus*	[110,159,160,161,162,163,164,165,166]
	Peroxidase and oxydoreductase	DAB, ATBS, OPD, pHQ, and others in the presence and absence of hydrogen peroxide	

* data obtained for Fab IgG, ** in the experiments, antibodies obtained on a sorbent with immobilized antigen were used. Abbreviations: AMC—7-amino-4-methylcoumarin, ATBS—2,2′-azino-bis-(3-ethylbenzothiozolin-6-sulfonic acid) diammonium salt, BApNA—benzoyl-l-arginine p-nitroanilide, BSA—bovine serum albumin, CSF—cerebrospinal fluid, DAB—3,3′-diaminobenzidine, HQ—hydroquinone, MBP—basic myelin protein, MCA—4-methylcoumaryl-7-amid, pNA—p-nitroanilide, OP—oligopeptides, OPD—o-phenylenediamine, VIP—vasoactive intestinal peptide.

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
