# Peer review of "Immunoglobulins with Non-Canonical Functions in Inflammatory and Autoimmune Disease States"

_ijms, 2020, doi:10.3390/ijms21155392_

Round 1

Reviewer 1 Report

This is a nice review of antibody responses. It is well written and I do not have any major objections aside from suggestions below.

  1. Though the review extensively covers microbiology of effects of antibodies, it appears that it does not explain the genetic architecture of antibodies which is related to their functions. Maybe I missed it, but I actually did not see the CDRs nor the framework regions mentioned once. Including an introduction t antibody function and sequence/structure would give context to rest of the work.
  2. Authors write a lot about the diversity of antibodies. This is a subject that is under a lot of investigation, mostly focusing on discerning the variability in CDRs, mostly by analyzing NGS outputs. Just like the introduction section on antibody structure/function that puts context to the whole review, including a few words on antibody diversity on repertoire level would allow the reader to get more context. For examples please see some recent reviews on the subject:
    1. https://www.tandfonline.com/doi/full/10.1080/19420862.2020.1729683
    2. https://www.ncbi.nlm.nih.gov/pmc/articles/PMC5826328/
    3. https://academic.oup.com/bib/article/doi/10.1093/bib/bbz095/5581643
  3. The review touches quite a lot on therapeutic viability of antibodies. However I have not seen any mentions of the drugs on the market, in clinical trials etc. Yearly reviews by Janice Reichert (Antibodies to Watch) are typically a good reference to provide therapeutic context.
  4. Figure 1D - you have 'paratop', I am pretty sure it should be 'paratope'.
  5. Figure 1 - please make Fc larger or more salient, it is hard to read.

Author Response

Replies to Reviewer 1

The authors are really grateful to Reviewer 1 for insightful review comments that significantly help improve the manuscript. Your comments allowed us to look at this review from a different perspective. We marked all adjustments in red.

Please note that your review comments are shown in italic below and our replies in non-italic.

On the comment of [1. Though the review extensively covers microbiology of effects of antibodies, it appears that it does not explain the genetic architecture of antibodies which is related to their functions. Maybe I missed it, but I actually did not see the CDRs nor the framework regions mentioned once. Including an introduction t antibody function and sequence/structure would give context to rest of the work.]

Reply: We agree that the inclusion of data on the structure of antibodies allows for a broader context and better description of the antibodies function. We have added a description of antibody genetics and architecture to the manuscript (please see lines 62-74).

On the comment of [2. Authors write a lot about the diversity of antibodies. This is a subject that is under a lot of investigation, mostly focusing on discerning the variability in CDRs, mostly by analyzing NGS outputs. Just like the introduction section on antibody structure/function that puts context to the whole review, including a few words on antibody diversity on repertoire level would allow the reader to get more context. For examples please see some recent reviews on the subject:…]

Reply: We thank Reviewer 1 for recommending data on the diversity of antibodies. This data allows you to better reveal the topic of the review. We have added data on the diversity of antibodies to the manuscript (please see lines 55-59).

On the comment of [3.  The review touches quite a lot on therapeutic viability of antibodies. However I have not seen any mentions of the drugs on the market, in clinical trials etc. Yearly reviews by Janice Reichert (Antibodies to Watch) are typically a good reference to provide therapeutic context.]

Reply: To provide a therapeutic context, we have added information on clinical therapeutic agents based on catalytic antibodies (please see lines 641-642 and 697-709).

On the comment of [4.  Figure 1D - you have 'paratop', I am pretty sure it should be 'paratope'.]

Reply: You are absolutely right. We've fixed this accidental literal error.

On the comment of [5.  Figure 1 - please make Fc larger or more salient, it is hard to read.]

Reply: We have corrected the Figure 1 and made it more readable.

Reviewer 2 Report

The authors have reviewed the roles of immunoglobulins, in particular, for their non-canonical function that is associated with catalytic antibodies. The authors systematically summarized the functions and types of catalytic antibodies in normal and pathological conditions and possible translational perspectives of knowledge about those antibodies. The manuscript is very well written, with updated references.

Please find my minor suggestions below.

  1. The reviewer suggests authors consider introducing the diversity of Abs before functions of Abs, thus switching subheadings 2 and 3. 
  2. It would be better if authors put subheadings 5 to 6 to under the subheading 4 since the following sections contain information about catalytic Abs.
  3. The reviewer suggests authors write 1-2 paragraphs of concluding remarks to the end of the manuscript. 
  4. The reviewer suggests authors write the implication of Ab or catalytic Ab in the COVID-10 pandemic in a section or a paragraph in the concluding remarks

Please find my minor suggestions below.

Missing definition of acronyms, -scFv, sIgA, SLE

Line 106 and 118, FcμR was defined twice.

Line 265, change “A – adaptive maturation of nucleophilic centers of Abs originating from the initial germline Ig genes;” to “A. Adaptive maturation of nucleophilic centers of Abs originating from the initial germline Ig genes;”

Table 3. reference 129Error! Bookmark not defined.

Line 401: Thus, not so.

Line 418: tripeptide (18 ?M)

Line 444: ?-casein [114] and

Line 455: following order: streptococcal infection

Line 473: и ?-amyloid [109]

Line 542: Based on the data presented in Chapter 5,

Line 573: As described in Chapter 6.2, Abs

Line 607: clinicians, not doctors

Author Response

Replies to Reviewer 2

The authors are deeply grateful to Reviewer 2 for a deep and thorough analysis of our manuscript. Your recommendations have significantly improved the manuscript and potentially increased its impact. We have made several important corrections to the manuscript, as well as corrected minor inaccuracies and grammatical errors. We marked all adjustments in red.

Please note that your review comments are shown in italic below and our replies in non-italic.

On the comment of [1.  The reviewer suggests authors consider introducing the diversity of Abs before functions of Abs, thus switching subheadings 2 and 3.]

Reply: We have carefully considered your proposal, but unfortunately came to the conclusion that switching subheadings 2 and 3 is not possible in this context. The main purpose of this review is to describe the various functions of antibodies. The chapter 3 about antibody diversity provides a better explanation of the diversity of antibody biological functions. Moving this chapter to the beginning shifts the focus of this review article in a different direction. Therefore, we would like to leave subheadings 2 and 3 as they are.

On the comment of [2.  It would be better if authors put subheadings 5 to 6 to under the subheading 4 since the following sections contain information about catalytic Abs.]

Reply: We fully agree with your proposal. We have renumbered the subheadings 5 to 6 in the manuscript.

On the comment of [3.  The reviewer suggests authors write 1-2 paragraphs of concluding remarks to the end of the manuscript.]

Reply: The authors gladly satisfied your wishes. We have added a Conclusion section to the manuscript (please see lines 755-765).

On the comment of [4.  The reviewer suggests authors write the implication of Ab or catalytic Ab in the COVID-10 pandemic in a section or a paragraph in the concluding remarks]

Reply: Fighting the COVID-19 pandemic is an important challenge for the global scientific community. Therefore, the authors agree with the proposal of Reviewer 2. We have added a paragraph on the prospects for the use of catalytic antibodies in the treatment of COVID-19 infection (please see lines 710-721).

On the comment of [Missing definition of acronyms, -scFv, sIgA, SLE]

Reply: The authors have fixed the missing definition of acronyms.

On the comment of [Line 106 and 118, FcμR was defined twice.]

Reply: The authors have fixed it.

On the comment of [Line 265, change “A – adaptive maturation of nucleophilic centers of Abs originating from the initial germline Ig genes;” to “A. Adaptive maturation of nucleophilic centers of Abs originating from the initial germline Ig genes;”]

Reply: We have made the appropriate changes to the caption for Figure 1.

On the comment of [Table 3. reference 129Error! Bookmark not defined.]

Reply: The authors carefully checked the manuscript and corrected errors in the references.

On the comment of [Line 401: Thus, not so.]

Reply: We fix it.

On the comments of [Line 418: tripeptide (18 ?M); Line 444: ?-casein [114] ; Line 473: и ?-amyloid [109]].

Reply: The authors have fixed the text formatting errors.

On the comment of [Line 455: following order: streptococcal infection ]

Reply: Yes of course. We have changed the order from highest to lowest.

On the comments of [Line 542: Based on the data presented in Chapter 5; Line 573: As described in Chapter 6.2, Abs ]

Reply: The authors carefully checked the manuscript and corrected errors when referencing various chapters.

On the comment of [Line 607: clinicians, not doctors.]

Reply: Thank you for suitable comment. The authors corrected the incorrect choice of the word.